# Filling the Gap: The Immune Therapeutic Armamentarium for Relapsed/Refractory Hodgkin Lymphoma

**DOI:** 10.3390/jcm11216574

**Published:** 2022-11-06

**Authors:** Esther Hazane Leroyer, Caroline Ziegler, Charline Moulin, Arnaud Campidelli, Caroline Jacquet, Marie Thérèse Rubio, Pierre Feugier, Simona Pagliuca

**Affiliations:** 1Service d’Hématologie Clinique, Hôpital Brabois, Centre Hospitalier Regional Universitaire de Nancy, 54500 Vandoeuvre les Nancy, France; 2CNRS UMR 7365 IMoPa, Biopole de l’Université de Lorraine, 54505 Vandoeuvre les Nancy, France; 3INSERM U1256 Nutrition-Génétique et Exposition aux Risques Environnementaux (NGERE), Université de Lorraine, 54506 Vandoeuvre les Nancy, France

**Keywords:** Hodgkin Lymphoma, immunotherapy, CAR-T cells, mechanisms of resistance, immune escape

## Abstract

Despite years of clinical progress which made Hodgkin lymphoma (HL) one of the most curable malignancies with conventional chemotherapy, refractoriness and recurrence may still affect up to 20–30% of patients. The revolution brought by the advent of immunotherapy in all kinds of neoplastic disorders is more than evident in this disease because anti-CD30 antibodies and checkpoint inhibitors have been able to rescue patients previously remaining without therapeutic options. Autologous hematopoietic cell transplantation still represents a significant step in the treatment algorithm for chemosensitive HL; however, the possibility to induce complete responses after allogeneic transplant procedures in patients receiving reduced-intensity conditioning regimens informs on its sensitivity to immunological control. Furthermore, the investigational application of adoptive T cell transfer therapies paves the way for future indications in this setting. Here, we seek to provide a fresh and up-to-date overview of the new immunotherapeutic agents dominating the scene of relapsed/refractory HL. In this optic, we will also review all the potential molecular mechanisms of tumor resistance, theoretically responsible for treatment failures, and we will discuss the place of allogeneic stem cell transplantation in the era of novel therapies.

## 1. Introduction

Hodgkin lymphoma (HL) is an unusual malignancy, belonging to Bcell neoplasms, characterized by exceptional epidemiological, pathophysiological and clinical features, including a bimodal incidence pattern, a highly inflammatory tumor environment characterized by a low number of malignant cells (Hodgkin and Reed-Stenberg cells -HRS), a high chemo and radiosensitivity [1].

The World Health Organization (WHO) recognizes two histological groups, differentiating distinct clinico-pathological entities [2]: (I) classic HL (cHL), characterized by HRS cells (expressing surface CD30), dispersed throughout a heterogenous inflammatory infiltrate, and constituting more than 90% of cHL cases; (II) nodular lymphocyte-predominant HL (NLPHL), in which malignant cells are aberrant CD20+ lymphocytes (lymphocyte predominant (LP) cells) lacking typical HRS features (CD15- and CD30-), representing about 5–10% of HL cases. cHL is itself divided into four histological subtypes, nodular sclerosis HL (NSHL), lymphocyte-rich HL (LRHL), mixed cellularity HL (MCHL), and lymphocyte-depleted HL (LDHL).

At disease presentation cHL usually infiltrates supradiaphragmatic nodal sites, spreading only in later phases to contiguous lymphonodes eventually with splenic or other extranodal involvements.

Treatment algorithms are based on the disease stage and are driven by medical imaging with response rates above 90% for early stages and about 80% for advanced diseases [3,4,5,6]. Nevertheless, relapsed/refractory (R/R) HL still represents a significant source of morbidity and mortality for a minor fraction of patients with this otherwise curable disease. To date, due to its pathophysiological background, a consistent framework of immunomodulating treatment strategies is available to overcome the failure of classical treatments. In addition, autologous and allogeneic hematopoietic cell transplantation have been occupying a relevant place in treating R/R HL since several decades. Last but not the least, other investigational cellular therapies have raised increased interest as potentially efficacious anti-CD30 targeting model [7,8,9,10,11,12,13,14,15].

This review focuses on the recent advancements of immunotherapy and cellular therapies in R/R HL, offering an up-to-date overview of investigational and approved strategies, highlighting in parallel the molecular mechanisms underlying escape, progression and treatment resistance in this dismal category of patients.

## 2. The Epidemiological, Immunogenetic and Environmental Background of Hodgkin Lymphoma

Epidemiologically, the burden of HL has been shown to vary across countries, impacted by sex, age, and past history of Epstein-Barr virus (EBV) infection, human immunodeficency virus (HIV), autoimmune diseases, exposure to pollution, smoking, family history and socio-economic status [16,17,18,19]. Data from the Global Cancer Observatory (GBO) [20] show that the global age-standardized incidence rate (ASR) of HL is 0.98 per 100,000 people, with, 83,087 new cases of Hodgkin lymphoma reported across the world in 2020 (Figure 1A,B). In particular, increased incidence is observed in high-economically developed areas, with rates of new diagnosis rising over the time, while high mortality is detected in low-income countries (Figure 1C,D). In historical cohorts, the probability of developing HL in young adults inversely correlated with the number of siblings, suggesting a causal link with the lack of early exposure to infectious agents, possibly affecting immune response homeostasis [21]. EBV infection is historically considered a major etiological factor for HL; however, EBV genome is found integrated in a limited number of HL cases and a causal role may be identified only in EBV-positive HL [22]. Age-related incidence rates rise steadily during childhood with a peak first at 20–25 (highest frequency in females), followed by a decrease until middle age before rising again reaching a second peak around age 75–79 (highest frequency in males, Figure 1B).

Genetic factors may influence the risk of HL. This was clear from the early 1990s when an increased incidence was observed in identical twins (99-fold compared to dizygous twins) [23]. More recent genome wide association analyses (GWAS) have demonstrated in the genetic orchestra predisposing to HL, the implication of a number of genes, with one of the strongest effects played by human leucocyte antigen (HLA) class I loci, followed by genes involved in germinal center reaction, T cell differentiation, *NF-κB* activation [24]. Variants in *IL13*, *GATA3*, *HLA* class II regulators, Polymorphic Killer cell immunoglobulin-like receptor (*KIR*) haplotypes of group B have been shown strongly associated with EBV-negative HL cases in young adults [24,25], whereas polymorphisms close to HLA class I loci were mostly associated with EBV-positive HL [26]. Metanalytic HLA imputations from GWAS data showed specific alleles strongly associated with the risk of EBV-positive HL including *A*01:01*, *C*07:01*, *B*08:01*, *DRB1*03:01*, *DQB1*02:01*, *B*37:01* [27].

## 3. Pathophysiological Background

### 3.1. Origin and Molecular Landscape of Malignant Cells in Hodgkin Lymphoma

HRS cells are supposed to stem from germinal center B-cells, albeit they do not harbour typical B cell lineage-markers [1]. These cells have been shown to share typical transcriptome patterns with CD30+ B cells and are characterized by a remarkable downregulation of genes regulating chromatin homeostasis and cytokinesis, explaining their genomic instability and bi or multinuclearity [28]. HRS cells are immunophenotypically characterised by overexpression of *CD30*, *CD15*, *MUM-1*, *PAX-5* [1].

CD30 is a member of tumor necrosis factor receptor (*TNFR*) superfamily (*TNFRSF8*), discovered for the first time in HRS cells [29,30]. The binding of CD30 to its ligand CD30L produces a trimerization of the receptor with a signal transduction which recruits TNFR-associated factors (TRAF), in particular TRAF2 and TRAF5, activating multiple downstream pathways, especially NF-KB [31]. CD30 is expressed on activated B, T and natural killer (NK) cells and on viral-infected cells. CD30-positive lymphocytes are generally located in the follicular areas of lymphoid organs [32]. Interestingly, increased CD30 expression can be also found in some autoimmune disorders, including systemic lupus erythematosus, inflammatory bowel diseases and rheumatoid arthritis [29,30]. Functionally, CD30 may promote different downstream signals (cell proliferation, survival or apoptosis) depending on the specific tissue and the target cells. CD30 plays a pleiotropic role at the hinge between T and B immune homeostasis, with a role both in T cell immune responses and regulation, proliferation and differentiation, as well as B-cell proliferation [33].

The B cell lineage was revealed by detection of rearranged immunoglobulin (Ig) genes (VH genes) and class-switch recombination in Hodgkin cells; and loss of classical B-cell phenotype is probably caused by epigenetic silencing of key regulators of B cell differentiation and upregulation of transcriptional antagonists during the process of lymphomagenesis [28,34].

A valuable source of information concerning pathophysiology and genomic landscape of HL derives from studies focusing on genomic and epigenetic landscape of enriched HRS cells. These cells are characterized by high genomic instability, including somatic mutations, translocations, copy number alterations of many immune genes, including Ig loci, T cell receptor (TCR) gamma, *JAK2*, *PD-L1* and *PD-L2* [35,36,37,38,39,40,41,42,43]. Recurrent genomic aberrations are particularly found in genes promoting cell proliferation, survival and immune escape pathways. In this context, different pathways have been identified as dysfunctional either because of genomic or transcriptional alterations. JAK/STAT pathway has been found constitutively activated in HRS cells as a result of inactivation of its negative regulators (such as SOCS1 and PTPN1) or aberrations activating STAT6 or JAK2 [36]. NF-kB pathway is also highly dysregulated through genomic aberrations altering the function of related genes (including *TNFAIP3*, *REL*, *NFKBIA*, *IKBKB*, *NFKBIE*) [35,44]. Somatic mutations can alter members of the NF-κB pathway both via gain of function hits (observed especially in *REL*, *MAP3K14*, *BCL3*), or through inactivating events of negative regulators such as *TNFAIP3*, *NFKBIA* and *NFKBIE* [45]. From studies of HL cell-lines it is now clear that the same HL-cell clones can present with different mutations in line with the need for a multi-hit process to dysregulate the NF-κB pathway [46].

PI3K-AKT pathway is also a key pathophysiological axis deregulated in HL. Somatic aberrations of *ITPKB*, may be responsible of constitutive AKT signalling and are detected in a consistent portion of patients. Similarly, *GNA13* loss has been shown able to lead impaired apoptosis, promoting lymphomagenesis in vivo [46,47]. *XPO1* a nuclear export receptor that mediates translocation of various RNAs and proteins from the nucleus into the cytoplasm has been also found mutated in many HL patients, with a highly selected genetic event (E571K) supposed to have a role in HL carcinogenesis [48]. Moreover, aberrant expression of *PDGFRA*, *DDR2*, *TRKA* and *TRKB* has been found in correlation with a constitutive activation of the PI3K-AKT pathway [1,49].

Recurrent gain of function genetic lesions of *PD-L1* and *PD-L2* genes, mostly associated with their overexpression, are consistently found across HL cohorts, with chromosome 9p24 often found amplified or polysomic. These alterations, when present at diagnosis, have also shown a negative prognostic impact in terms of progression free survival (PFS). When overexpressed in HRS-cells, they inhibit the activity of cytotoxic T cells and other PD-1 expressing cells, contributing to generate an immunosuppressive environment with exhaustion of T cell effectors [37,42,50].

Other molecular mechanisms involved in HL lymphomagenesis include the overexpression of several receptor tyrosine kinases (RTKs) found upmodulated in up to 75% of cHL cases, including *PDGFRA*, *DDR1*, *DDR2*, *EPHB1*, *TRKA*, *RON*, *CSF1R*, and *MET* [28] Besides alteration of these proliferation and differentiation axes, another fundamental molecular dysfunction at the bases of HRS persistence and HL progression includes somatic aberrations in *HLA* class I related antigen presentation, via the genomic events impairing *B2M* gene, causing decrease in presentation of neoantigen to CD8 + T cells and favoring immune escape mechanisms [35] Other dysregulations of the antigen presentation machinery in HL involve the class II transactivator *CIITA*, resulting in impairment of presentation of class II-restricted antigens to T-helper responses [34,35]

In case of EBV positive HRS-cells, EBV latent membrane proteins LMP1 and LMP2A provide infected B cells with growth and survival signals, molecularly mimicking the activation of CD40 receptor and activating the NF-κB signalling pathway as well as BCR signalling, resulting in HRS-cell survival [1,51,52].

### 3.2. Microenvironmental Interactions

The immune/inflammatory microenvironment surrounding HRS-cells present unique characteristics and sustains proliferation and survival signalling, facilitating at the same time the escape from adaptive immune surveillance.

The HL specific tumor microenvironment (TME) consists mainly of CD4+ T cells placed around HRS-cells, CD4+—T regulatory (T-reg) cells, macrophages, eosinophils, mast cells, NK cells, fibroblasts. The interactions between HRS-cells and the TME act through the autocrine and paracrine effects of cytokines and chemokines, modulating the activity of surrounding cells [53,54]. Specifically, CCL5, CCL22 and CCL17 (produced, respectively, by regular T cells, thymus and activation-related T cells, macrophages) have been shown to attract CCR4 expressing mast cells, Th2 CD4+ and T-reg cells in the peri HRS-cell milieu [54,55,56,57,58]. The implication of macrophages in supporting tumor proliferation has been also demonstrated in HL and secretion of macrophage migration inhibitory factor (MIF) may contribute to the proliferation of HRS-cells [53,59,60]. Expression of IL-3 mediating the production of TNF-α allows the activation and proliferation of fibroblasts [57]. Fibroblasts may also attract eosinophils and Th2 cells through secretion of CCL11 [61].

If genetic lesions of HRS cells alter intrinsic proliferation and differentiation pathways, dysfunctional signals essential to disease ontogenesis are provided by the microenvironment. For instance, activation of NF-κB pathway originates from the expression of *CD30*, *CD40* or TNFR members, such as *TNFRSF1A* or *TNFR11A*, found on the surface of HRS cells [45,46,53]. The TME cells secrete CD30 ligand (CD30L), CD40L and TNF-alpha, which bind to their respective receptors and trigger the signalling cascade. The JAK-STAT constitutive activity is enhanced by signalling of IL-13, which engages the IL-13 receptor and leads to HRS cell proliferation and survival by phosphorylated STAT6 [36]. Overall, the interactions are multiple and diverse and allow for the proliferation and survival of HRS-cells through a supportive and stimulating milieu, influenced itself by the expansion of HRS cells. Production of CCL17 promotes differentiation capacities of CD4 + T cells towards an immunosuppressive T-reg phenotype [58,62,63]. T-reg cells and HRS cells produce also IL-10 and TGF-ẞ which inhibit cytotoxic functions of T cell CD8+ effectors [64,65] Moreover, there is evidence for overexpression of surface molecule maintaining peripheral tolerance to tumor neoantigens impregnating the polyclonal lymphocyte infiltrate peri-HRS cells. As in other neoplastic disorders, the overexpression PD-L1 and PD-L2 bind to PD-1 receptor, notably expressed on T cells, modifies T cell activity, contributing to T cell exhaustion and immune suppression [50,66]. The effect on immune tolerance is also enhanced by the loss of expression of HLA-class I and II on HRS causing impaired CD8+ and CD4+ recognition [46,67].

### 3.3. Mechanisms of Resistance

The pathophysiological mechanisms of resistance in R/R HL are largely underexplored because of the lack of systematic studies comparing diagnosis vs. relapsed/progressed diseases.

Although new targeted therapies (described in detail below) significantly improved the overall survival (OS) of R/R HL patients, the relapse mechanisms are still unclear.

While anti-CD30 therapies brought solid hope to patients who relapsed after an autologous stem cell transplant, resistance mechanisms have been observed [68,69]. In case of BV, an evoked resistance mechanism was the upregulation of the multidrug resistance gene MDR1 demonstrated in vitro on HL-cell lines. The addition of cyclosporine A (CSA), an inhibitor of MDR1 [70], restored sensitivity to ant-CD30 in resistant HL-cell lines from mice. This observation motivated a phase I trial in R/R HL patients (*n* = 14) receiving CSA and anti-CD30 showing a promising overall response rate (ORR) of 75% and complete remission (CR) in 42% of patients. [70,71] Intriguingly loss of CD30 after anti-CD30 exposure seems to be a more frequent pathological event for cutaneous and anaplastic large cell lymphoma than for HL [72,73,74].

In 2006, the Spanish Hodgkin Lymphoma Study Group analyzed the gene expression profile of anatomopathological samples from lymph nodes at diagnosis and retrospectively correlated it to treatment outcome. Genes expressed by specific subpopulations of T cells (e.g., *CD8B1*, *CD3D*, *CD26*, *SH2D1A*), macrophages (e.g., *ALDH1A1*, *LYZ*, *STAT1*), and plasmacytoid dendritic cells (e.g., *ITM2A*) were associated with unfavorable outcome, supporting the idea that the tumor microenvironment plays a major role in treatment failure [75]. Tumor associated macrophages and monocytes significantly associated with primary treatment failure and with an increased likelihood of relapse after autologous hematopoietic stem-cell transplantation. In immunohistochemical analyses, the number of CD68+ tumor-infiltrating macrophages correlated with shortened PFS [59,76]. Furthermore, the percentage of cytotoxic-T cells in Hodgkin Lymphoma is significantly associated with poor outcome and identified as an independent biological prognostic marker [76].

One of the possible mechanisms of refractoriness after anti-PD-1 therapy could be related to the compensatory expression of other non-redundant checkpoint negative regulators. For instance, analysis of the microenvironmental niche in cHL with immunofluorescence microscopy of tumor biopsies, enlightened the presence of a markedly expanded population of CTLA-4–positive, PD-1–negative, and LAG-3–negative T cells [77,78]. These CTLA-4–positive T cells were accompanied in tumors from patients previously treated with PD-1 inhibitors and in some cases after allo-HCT, by enhanced expression of CD86 (a ligand for CTLA-4) on HRS cells and TAMs, contributed to treatment failure.

## 4. Management of Relapsed/Refractory Hodgkin Lymphoma

After a stage-adapted standardized first-line therapy (discussed in detail elsewhere) [52,79,80,81] up to 25% of patients may eventually experience refractoriness or disease relapse [52]. For these subjects, a second-line treatment should always be considered and optimized based on age, comorbidities and patient’s status.

High dose chemotherapy followed by autologous stem cell transplantation (ASCT) occupies a pivotal place essentially after failure of first-line treatment, with the possibility of achieving high CR rates (especially in relapsed more than primary refractory patients).

To date, no study has shown the superiority of a specific salvage regimen above others in the context of R/R HL and the main take home messages from real-life experiences is to always consider previous regimens, cumulative dose of specific agents, the need to harvest hematopoietic stem cells (HSCs). In 2013 the Lymphoma Study Association (LYSA) provided guidelines for selecting transplant eligible patients. [82] A risk adapted strategy was conceived based on three risk factors at progression: primary refractory disease, remission duration <1 year, stage III/IV at disease relapse. Based on these recommendations, patients may be divided into three risk categories: (I) high-risk: refractory or early recurrence and stage III/IV; (II) intermediate risk: early relapse or stage III/IV; (III) standard risk: patients relapsing without risk factors [82]. Generally, in fit patients a ASCT is recommended in all these situations. Still under discussion, especially in the post-anti-CD30 and checkpoint inhibitors era, is the place of allogeneic hematopoietic cell transplantation (allo-HCT), which seems to improve PFS but remains associated with high level of toxicities [83]. Indeed, today in R/R HL the spectrum of possible therapeutic options covers a wide range of therapeutics that may let replace the role of allo-HCT. Hereafter, we attempt to gather the most updated literature on the current approved and investigational treatments in R/R HL.

### 4.1. The Evolving Place of Brentuximab Vedotin

Brentuximab vedotin (BV) is a highly effective antibody-drug conjugate that combines a CD30-specific monoclonal antibody with monomethyl auristatin E (MMAE), an antimitotic agent (vedotin). Its role in the therapeutic pipeline of HL has changed over the time contributing to salvage, consolidation and first-line treatment combinations. It was first approved by the Food and Drug administration (FDA) in 2012 for use as salvage therapy in R/R HL after the results of a phase II trial showing an overall response rate (ORR) of 75% in heavily pretreated patients [84,85]. It was subsequently approved for post-ASCT maintenance for high-risk patients based on the phase III AETHERA study, showing an improved 5-year PFS after BV consolidation compared to placebo [86]. In 2018, after the results of ECHELON-1 trial BV gained the indication in first-line treatment for patients with previously untreated advanced stage HL in combination with doxorubicin, vinblastine, and dacarbazine (A + AVD) [87].

Based on encouraging results in the setting of post-ASCT relapses, BV was then studied in pre-transplant salvage contexts. A phase II trial exploring its role as second-line treatment before high dose chemotherapy and ASCT showed ORR over 60% with CR rates of 35%. Importantly, in this study, almost 50% of the patients were in partial remission (PR) or CR after BV alone without additional combination chemotherapy, before receiving intensification and ASCT, paving the way for the possibility of less toxic pre-transplant salvage regimens [88]. BV as a single agent was also evaluated in the context of positron-emission tomography (PET)-guided approaches, with satisfactory outcomes and the possibility of sparing more toxic chemotherapy in a consistent fraction of patients [89,90].

Many different BV-based combination schemas have also been investigated. An effective drug often studied in association with BV is bendamustine. A large multi-center phase I/II trial studied the efficacy of BV in combination with bendamustine, followed by optional ASCT and additional BV maintenance. CR was 74% with ORR reaching 93% and 2-year PFS of 63% overall and 70% in ASCT group [91]. Other prospective or retrospective studies offering the same combination, reported ORR between 70 to 100% [92,93,94,95].

BV has been also studied in the setting of several chemotherapy combination schemes, increasing the likelihood of achieving pre-ASCT CR; however, at the expense of cumulative toxicities. For instance, various studied have explored in R/R HL the association with ifosfamide, carboplatin and etoposide (ICE) showing excellent outcomes (ORR > 80% and CR > 70%), however with high rates of grade III/IV neutropenia and thrombopenia and in some cases the occurrence of multiorgan failure [89,96,97,98]. Similarly, the combination with dexamethasone, cisplatin, and cytarabine (DHAP) and etoposide, methylprednisolone, cytarabine and cisplatin (ESHAP) have been experimented, showing very good survival and response outcomes but with high rates of severe cytopenias, febrile neutropenia, liver and renal failures [99,100].

### 4.2. Targeting the PD-1 Axis

Preclinical studies demonstrating the pathophysiological relevance of the PD-1 pathways in the context of HL have paved the way to integrate immune checkpoint inhibitors, specifically nivolumab and pembrolizumab, in the current treatment pipelines.

#### 4.2.1. Nivolumab

Nivolumab is a fully humanized IgG4 anti-PD-1 blocker targeting epitopes on PD-1 and impeding the interaction with PD-1L and PD-2L [101]. It was first evaluated in R/R HL context, in general in patients receiving at least >3 lines of treatment and ASCT. Results of first phase I and II trials as well as retrospective trials showed ORR ranging from 60–70% and CR of around 20–30% [102,103,104,105]. The CheckMate205 study whose results were presented in 2016, investigated the use of this agent in 80 patients with R/R HL after failure of ASCT and/or BV, showing an objective response in 66% of patients with low frequent severe adverse events including grade III/IV neutropenia and pancreatitis (in 5%) [106]. The extended follow-up study of this multi-center single arm trial (243 patients) continued to demonstrate frequent and durable responses with a favorable safety profile [105]. In real-life and retrospective cases series Nivolumab has demonstrated the concrete possibility to achieve alone CR, allowing treatment discontinuation [102,103,104,105]. Results of a long-term study based on the Early French Access Program showed a CR rate of almost 40% in heavily pre-treated R/R context, with 25% of patients in remission without further consolidation therapies, after a median of 20 cycles of Nivolumab, with a PFS and OS of 48% and 87%, respectively [104]. Of note, in this study, a group of patients received anti-PD-1 agents after allogenic HCT (allo-HCT, 37%), underscoring the feasibility of anti-PD-1 also in this dismal salvage context [104].

BV-Nivolumab associations have been studied in the setting of second-line R/R HL, displaying very interesting results. A phase I/II trial showed excellent three-year outcomes demonstrating ORR of up to 87%, CR of 67% and PFS of 77% after a total of 4 cycles, with patients proceeding to ASCT presenting with the best outcomes (91% of 3-year PFS) [107]. Despite frequent infusion-related reaction (43%) and mild immune-allergic manifestations (including pneumonitis and rash in 18% of patients, this combination was well tolerated, with no patient experiencing treatment discontinuation. A recent phase II trial explored the combination of Nivolumab with ICE regimen (NICE) in 9 patients showing above 90% of complete remission rates at the end of the program, with manageable toxicity [108].

#### 4.2.2. Pembrolizumab

Pembrolizumab is another monoclonal antibody inhibiting PD-1, licensed for cancer immunotherapy, FDA-approved in 2020 for R/R HL. This approval was based on the results of KEYNOTE-204 trial a phase 3 randomized trial (pembrolizumab vs. BV) in 304 patients with R/R HL, after failure of at least one multi-agent regimen. PFS was significantly longer in pembrolizumab arm (median 13.2 vs. 8.3 months in BV group) despite frequent adverse events (seen in up to 30% of the patients receiving the anti-PD-1 agent, including pneumonitis, myocarditis, acute kidney injury, hyperthermia, febrile neutropenia, sepsis, musculoskeletal pain, diarrhea, rash) [109]. As matter of fact, the previous phase II KEYNOTE-087 trial offered already a clinical benchmark favorable to the use of pembrolizumab in the R/R setting, showing a two-year ORR of 72% and a CR of 28% [110]. A post-hoc analysis of a subset of 71 patients with primary refractory patients who relapsed after salvage ASCT showed overall an ORR of 87% and a CR of 35% also in this dismal category of patients. In terms of combinations pembrolizumab was investigated in association with gemcitabine, vinorelbine, liposomal doxorubicin (pembro-GVD) in a phase II trial in the setting of second-line therapy. After 2–4 cycles in transplant eligible patients ORR was 100% with a CR of 95%, with the majority of patients only requiring 2 cycles of pembro-GVD. Despite many adverse events of mild intensity (rash, hyperthyroidism, transaminitis, neutropenia), grade III/IV toxicity remained limited [111]. Experiences of associations of BV with pembrolizumab are very limited, and include a monocentric experience from the Belgian group on 10 patients receiving three cycles of this combination as bridge to transplant without major toxicity, and experiencing an ORR of 90% and a CR of 80% prior to consolidating ASCT [112]. A recent phase II trial investigated for the first time pembrolizumab as upfront first-line treatment in a sequential scheme followed by AVD. After three cycles of pembrolizumab 37% of patients were in CR with all patients achieving at least a PR, including in advanced stages, after 2 cycles of AVD [113]. Long-term results confirmed excellent PFS and OS which remained both at 100% after three years from therapy [114]. Despite the advent of novel therapies such as BV and immunotherapy for treatment of R/R HL gives promising results, the optimal treatment strategy for R/R HL remains unclear in the absence of comparative trials. A summary of FDA approved drugs for HL is shown in Figure 2.

#### 4.2.3. Other Investigational Anti-PD-1 Inhibitors

Sintilimab is a highly selective PD-1 antagonist whose efficacy was investigated in a phase II multicenter Chinese study in the context of R/R and heavily pretreated HL, revealing an excellent ORR of 80.4% although with high frequency of adverse event (any grade in 93% of the patients, grade III/IV in 18% of the cases). This compound was able to achieve CR in a patient with human immunodeficiency virus (HIV) associated HL refractory to two lines of treatment, without severe toxicity [115].

Camrelizumab is another anti-PD-1 drug that showed interesting response rates in a phase II trial including HL patients failing or ineligible to ASCT, with ORR of 76.0%, median PFS of 22 months and 3-year OS of 82.7% [116,117]. This agent was also studied in combination with low doses of decitabine, an hypometilating agent, known to improve T cell functions and to overcome a possible epigenetic-related overexpression of PD-1 [118,119], in a randomized phase II trial, demonstrating improved CR rates (71%) in patients with r/r cHL compared with camrelizumab monotherapy (32%). Further studies confirmed the efficacy of this combined therapy, suggesting a high synergistic antitumor activity of this combinatorial schema [120,121].

Penpulimab is a IgG1 anti-PD-1 agent that revealed clinical efficacy through a phase I/II trial, showing in 94 patients enrolled, an ORR of 89% with CR of 47% and excellent 1-year PFS and OS (respectively, 72% and 100%, however again with frequent toxicity (97% of patients experienced at least one adverse event) [122].

Tislelizumab is a IgG4 anti-PD-1 monoclonal antibody, studied, in R/R HL patients failing or ineligible to ASCT, in a phase II trial in China. ORR was of 87% and CR of 62.9% after a short follow-up (less than 12 months) [123].

Zimberelimab (GLS-010) is a last generation PD-1 blocker, which also recently showed its validity in R/R HL setting, based on the results of another multicenter phase II study conducted in China. For a median follow-up of 15 months, and with 85 patients enrolled, ORR was 90% and CR 32%, still with adverse event occurring in most of the participant (92%), and grade III/IV observed in 28% of the trial population [123].

### 4.3. Novel Agents Potentially Effective in R/R HL

Although conventional chemotherapy, BV and immune checkpoint inhibitors have populated the therapeutic scenario of R/R HL of highly efficient regimens, chemo-refractory and heavily pre-treated patients still may develop additional mechanisms of resistance, rendering most of the available options ineffective or extremely toxic. Indeed, a flourishing pipeline of agents and regimens is under investigation to offer to this very risky category of patients better survival outcomes and tolerance.

Selected agents and their interventional settings are listed in Table 1.

#### 4.3.1. Role of Allogeneic Stem Cell Transplantation

The place of allo-HCT is still of actuality in R/R HL, for its effectiveness in disease control through graft-versus-lymphoma (GvL) effect. However, the optimal timing and ideal group of patients for this procedure remain a matter of debate, with a major tendency to reserve this procedure after ASCT either as tandem procedure in high-risk patients or after failure of autologous rescue. The literature focusing on allo-HCT in HL is almost exclusively based on retrospective studies [124,125], highlighting the limited investigational potential invested in understanding the role of allo-HCT compared to other approaches. However, even in the absence of randomized clinical trials, allo-HCT either from an HLA identical sibling or from a matched unrelated or an alternative donor construes an attractive route for young patients after salvage regimens. The Spanish Lymphoma group (GELTAMO) together with the European blood and marrow transplant group (EBMT) conducted one of the largest phase II studies in this setting including 78 patients with R/R HL, receiving a matched related (MRD) or unrelated (MUD) or one antigen mismatched (MMUD) allo-HCT. Half of the patients were transplanted in CR or PR. Conditioning regimen was of reduced intensity and comported fludarabine and melphalan with or without anti-thymocyte globuline. Four-year PFS and OS were, respectively, 24 and 43% with chronic graft versus host disease (GvHD) inversely correlated with the risk of relapse, as demonstration of the disease control operated by the GvL effect [126].

An historical study on behalf of EBMT compared the outcomes in patients receiving reduced intensity (RIC) vs. myeloablative regimens (MAC), in a retrospective series of 168 patients, demonstrating the superiority of RIC in terms of non relapse mortality (NRM) and OS, with again the impact of chronic GvHD in decreasing the incidence of relapse [127]. A large retrospective study, including 285 patients receiving a RIC allo-HCT have investigated the factors associated with outcomes and identified chemo-refractory disease, poor performance status, age >45 and transplantation before 2002 as factors impacting NRM. This study was also concordant in showing the chronic GvHD benefit in terms of risk of relapse [128]. A retrospective historical comparison of allo-HCT vs. conventional therapy based on the availability of a donor in 185 patients failing ASCT (122 with a donor, 63 without a donor), showed the superiority in terms of PFS, and OS in the transplanted group [129].

Another historical series informed on the outcomes after unrelated matched allo-HCT based on data from the Center for International Blood and Marrow Transplant Research (CIBMTR). In this study TRM was 33% at 2-years; whereas 2-years PFS and OS were, respectively, 20 and 37% [130].

Auto-allo tandem approaches have been recently assessed and offer interesting outcomes. In 126 patients (41% of which transplanted in active disease), PFS was 53%, OS 73% and cumulative incidence (CI) of relapse was 34%, with acceptable toxicity and a NRM of 13% [131]. A less recent study evaluated the outcomes of transplant-based consolidation therapies after 4 cycles of BV, with 12 patients receiving allogeneic HCT.

**Table 1 jcm-11-06574-t001:** Selected phase II and III studies for investigational agents in relapsed/refractory Hodgkin Lymphoma.

Agent	Mechanism of Action	NCT	Phase	Study Design	Number of Therapeutic Lines	Start Date/Status	Reference	Results if Available
Anti-PD-1/PD-L1								
PENPULIMAB	Monoclonal antibody (anti-PD-1)	NCT05244642	III	Randomized, Open, Multi-center Phase III Study to Evaluate the Efficacy and Safety of Penpulimab Monotherapy vs. Standard Chemotherapy Selected by Investigator	≥2, relapsed after ASCT	February 2022/Recruiting		Not available
CAMRELIZUMAB (SHR-1210) vs. Chemotherapy	Monoclonal antibody (anti-PD-1)	NCT04342936	III	Open-label, multicenter, randomized trial to evaluate the efficacy of Camrelizumab monotherapy or chemotherapy	≥2, relapsed after ASCT	July 2020/Recruiting		Not available
TISLELIZUMAB	Monoclonal antibody (anti-PD-1)	NCT03209973	II	Single Arm, Multicenter, Phase 2 Study of BGB-A317 as Monotherapy in R/R cHL	Relapsed after ASCT	July 2017/Completed	Song et al. Leukemia 2020 [123]	ORR:87%; CR: 63%
TISLELIZUMAB	Monoclonal antibody (anti-PD-1)	NCT04318080	II	Multicenter, Open-Label Study to evaluate the efficacy of Tislelizumab (BGB-A317) in Patients With Relapsed or Refractory Classical Hodgkin Lymphoma	Relapsed after ASCT	August 2020/Active Not recruiting		Not available
TISLELIZUMAB	Monoclonal antibody (anti-PD-1)	NCT04486391	III	Multicenter, open-Label, randomized Controlled Phase 3 Study of Tislelizumab Monotherapy Versus Salvage Chemotherapy	≥2, relapsed after ASCT	September 2020/Recruiting		Not available
**Immunotherapy associations**								
IPILIMUMAB +/− Nivolumab	Monoclonal antibodies (anti-CTLA-4, anti-PD-1)	NCT04938232	II	Open-label, multi-cohort, multi-center of ipilimumab with or without nivolumab.	≥2 including PD-1 monoclonal antibody, relapsed after ASCT	September 2021/Recruiting		Not available
Brentuximab Vedotin and Nivolumab with or without IPILIMUMAB	Monoclonal antibodies (anti-CD-30, anti-PD-1, anti-CTLA4)	NCT01896999	I/II	Randomized Phase II Study of the Combinations of Ipilimumab, Nivolumab and Brentuximab Vedotin.	≥1	July 2013/Recruiting	Diefenbach et al. Lancet Haematol 2020 [132]	*n* = 64; (I) BV + Ipilimumab: ORR= 76%; CR = 57%; (II) BV+ nivolumab ORR = 82%; CR = 61%; (III) BV + Ipilimumab + nivolumab: ORR = 82%; CR = 73%
MAGROLIMAB and Pembrolizumab	Monoclonal antibodies (anti-CD47, anti-PD-1)	NCT04788043	II	A Phase 2 Study of Magrolimab and Pembrolizumab	≥2	June 2022/Recruiting		Not available
CAMRELIZUMAB (SHR-1210) Alone or in Combination with Decitabine	Monoclonal antibody (anti-PD-1)	NCT03250962	II	Multicohort, decitabine-plus-SHR1210 single-arm clinical trial. Evaluate the long-term response duration with decitabine-plus-SHR-1210	≥4, ≥3 months from ASCT	September 2017/Recruiting	Liu et al. J Immunother Cancer 2021 [133]	*n* = 61; I) SHR-1210: CR= 32%; II) SHR-1210 + decitabine: CR = 79%
CAMRELIZUMAB (SHR-1210) Combined With GEMOX	Monoclonal antibody (anti-PD-1)	NCT04239170	II, III	Open-label, single arm, Phase 2 study to evaluate efficacy and safety of PD-1 inhibitor Camrelizumab(SHR-1210) combined with Gemox who will receive ASCT	≥3, relapsed after ASCT	January 2020/Recruiting		Not available
Nivolumab with RUXOLITINIB	Monoclonal antibody (anti-PD-1), JAK2 inhibitor	NCT03681561	II	Multicenter, open-label, dose escalation/dose-expansion study to evaluate the tolerability, safety, and the maximum tolerated dose (MTD) of ruxolitinib when given with fixed dose nivolumab	≥2, including check point inhibitors	13 September 2018/Recruiting		Not available
**Bispecific Antibody**								
AZD7789	Bispecific Antibody (Anti-PD-1 and Anti-TIM-3)	NCT05216835	I/II	Open-label, Multi-center Study to Assess Safety, Tolerability, Pharmacokinetics and Preliminary Efficacy of AZD778	≥2, no previous treatment with anti-TIM-3	18 March 2022/Recruiting		Not available
				Phase 1 (Part A) Dose Escalation and Phase 2 (Part B) Dose Expansion				
**Other immunomodulating agents**								
ITACITINIB (INCB039110) and EVEROLIMUS (Afinitor)	JAK 1 inhibitor; mTOR inhibitor	NCT03697408	I/II	Open-label, single-group, study of itacitinib in combination with everolimus	≥2, relapsed after ASCT	11 February 2019/Recruiting		Not available
**CAR T-cells**								
CART30 cells	Autologous CART-30 cells	NCT02259556	I/II	CD30-directed Chimeric Antigen Receptor T (CART30) Therapy	≥2 or relapsed after ASCT	October 2014/Recruiting		Not available
HSP-CAR30	Autologous CART-30 cells	NCT04653649	I/IIa	Interventional, single arm, open label, treatment study to evaluate the safety, tolerability and efficacy of HSP-CAR30	Relapsed after ASCT who have received anti PD-L1 or Brentuximab; or Primarily refractory patients who do not reach CR after rescue	September 2020/Recruiting	Caballero et al., EHA 2022	Preliminary results: *n* = 11; ORR = 100%; CR = 62%
ATLCAR.CD30 cells	Autologous CART-30 cells	NCT02690545	Ib/II	Establish a safe dose of ATLCAR.CD30 cells to infuse after lymphodepleting chemotherapy and evaluate relative toxicities	≥2, CD30+ disease	August 26, 2016/Recruiting	Ramos et al. JCO 2020 [134]	*n* = 41; CRS = 26%; ORR = 72%; CR = 59%; 1-year PFS = 36%; 1-year OS = 94%

Interestingly, in this subgroup none of the patients relapsed or died showing the feasibility of this intensification with high rate of responses after BV. Of note 4 patients included in this series underwent BV- allo-HCT in the context of relapse after a first allogeneic procedure [134] A very recent case series examined the role of allo-HCT as first transplant procedure in 190 patients of which 63% were previously exposed to checkpoint inhibitors and BV. The 3-year NRM and relapse rate were 21% and 38%, respectively. After a median follow-up of 58 months, 3-year OS and PFS were 58% and 41%, respectively.

As to alternative donors, more recent study cohorts highlighted better outcomes, possibly for the constant improvement of pre-transplant management (use of BV associated with less cytotoxic regimens) and because of the use of post-transplant cyclophosphamide (PT-Cy) in haploidentical setting. A multicenter retrospective study revealed in 198 patients (133 receiving an HLA matched and 65 a haploidentical donor) a superior PFS (63% vs. 37%, *p* = 0.03) and an inferior cumulative incidence of relapse (24% vs. 44%, *p* = 0.008) in haploidentical vs. identical group, showing no differences in OS and NRM. A joint EBMT-CIBMTR-Eurocord studies reported the outcomes of 740 lymphoma patients (of which 283 HL) receiving an alternative donor transplantation. This study showed the superiority of haploidentical peripheral blood (PB) and bone marrow (BM) HCT compared to umbilical cord blood transplantation (UCBT). In this study the 4-year OS and PFS were 49 and 36% after UCBT, 58% and 46% after haploidentical BM and 59% and 52% after haploidentical PB, respectively [135]. Another large multicenter retrospective study on behalf of the lymphoma working party (LWP) of the EBMT evaluated 709 HL patients (98 haploidentical, 338 siblings, 273 MUD) and showed similar survival outcomes in the three groups, with haploidentical transplant being associated with lower risk of chronic GvHD than MUD. Genova group showed appealing results with a modified haploidentical approach (including unmanipulated BM graft, non-myeloablative conditioning, PT-Cy at day + 3 and +5, with cyclosporine start at day 0 and mycophenolate from day + 1) in 26 patients [136]. All patients received a previous ASCT and 65 had active disease at the time of haploidentical transplant. In this cohort, with a median follow-up of 24 months, the 3-year OS was 77%, and the 3-year PFS was 63%. Incidences of grade II-IV acute GvHD and chronic GvHD were 24 and 8%, respectively [136]. While haploidentical donor transplant platforms keep gaining a valid place in HL eligible to allo-HCT such that an open question remains which donor to select in case of availability of both MMUD (or even MUD) and haploidentical, more limited is the place of unrelated UCBT. Results of a study from Eurocord and EBMT including 131 adults with HL receiving unrelated UCBT demonstrated 4-year PFS and OS of, respectively, of 26% and 46%, with relapse incidence remaining high (44%) and NRM of 31% at 4 years [137].

#### 4.3.2. The Dilemma of Immune Checkpoint Inhibitors before Allo-HCT

Due to the consequences of an enhanced T cell activation, the immunological effects of the anti-PD-1 have raised some concerns as to their safety profile in the pre allo-HCT context, because of the potential risk of graft-versus- host disease (GVHD) and other immuno-allergic manifestations. The retained hypothesis is that the long half-life of nivolumab and pembrolizumab in the plasma would induce a persistent T cell activation of donor-derived immune cells increasing the risk of severe acute and chronic GvHD, immune-mediated pneumonia and other organ-related immune alterations in the recipient. Some translational data have attempted to explain the pathophysiology of these complications and biological analyses of a multicenter retrospective study conducted on 39 patients who received PD-1 blockade before allo-HCT have shown a significant reduction in PD-1 T cells and decreased Tregs as compared with the historical cohort of patients who did not receive checkpoint inhibitors [138]. An interesting study conducted by the Spanish group from the University of Barcelona, showed that nivolumab may persist in the blood until 56 days after transplant and that while patients receiving only calcineurin inhibitors had an increased incidence of severe acute GvHD and a more effector T cell profile, patients undergoing PT-Cy had a similar risk of GVHD and similar T cell profile to those without previous nivolumab exposure [139].

Available clinical data, deriving also mostly from retrospective series with limited number of patients, have shown elevated risk of acute 2–4 GvHD in patients receiving standard immunosuppressive regimens (around 30–40% in the first year post-HCT), with OS and PFS around 70% at three years [138,140]. A recent study analyzed instead outcomes after PT-Cy utilization comparing patients exposed to ICIs and patients receiving chemotherapy only [141]. In this experience from John Hopkins, no differences in terms of acute or chronic GvHD were seen between ICI (*n* = 37) and no-ICI groups (*n* = 68) with incidence of acute grade 2–4 GvHD, respectively, of 33% vs. 17% and of chronic GvHD of 3% vs. 14%. In this study, 3-year PFS and OS were slightly higher in ICI group as compared to the no-ICI group (90 and 94% vs. 65 and 78%, respectively), with incidence of lymphoma relapse ≤10% in both groups. It is noteworthy to mention that bone marrow was the preferred stem cell source in most of these studies and that this choice could have contributed to better outcomes and reduced incidence of immune-related complications. Altogether, these results are too limited to state on the right timing of allo-HCT after PD-1 blockade but they may alert on the need for a more efficient T cell depletion also in matched contexts and on the promising role of PT-Cy in this direction [142]. Table 2 summarizes principal studies analyzing allo-HCT outcomes in case of previous anti-PD-1 exposure.

### 4.4. Chimeric Antigen Receptor T Cells in Hodgkin Lymphoma

Chimeric antigen receptor (CAR) T cells have become today the paradigm of immunotherapy, able to direct the patient’s own immune cells against target cancer antigens, through an engineered receptor structure [143,144]. CAR T cells have demonstrated exceptional results in trials and real-life experiences for non-Hodgkin lymphoma, multiple myeloma and B cell acute lymphoblastic leukemia even in refractory or heavily pretreated contexts, showing, in many cases, complete and durable responses [145]. This success in other lymphoproliferative disorders and hematologic malignancies has encouraged investigations in the context of R/R HL [146]. Early cellular therapy approaches in HL were focused on targeting EBV-related antigens in the setting of 20–30% of EBV positive HL.

**Table 2 jcm-11-06574-t002:** Principal studies including patients exposed to immune checkpoint inhibitors before allo-HCT for R/R HL.

Design	Anti-PD-1 Molecule	Disease	N# of Patients Receiving Anti-PD-1 before Allo-HCT	Median Time of Anti-PD-1 before Transplant	GvHD Prophylaxis Type	Graft Source	II-IV Grade Acute GvHD	Chronic GvHD	OS	Relapse	NRM	Translational Findings if Available	Reference
Retrospective	Nivolumab and Pembrolizumab	HL/NHL	39	62 (7–260)	Heterogeneous	BM and PBSC	44%	41%	89% at 1 year	14 at 1 year	11 at 1 year	depletion of PD-1+ T cells and reduction in T-reg cells	Merryman et al. Blood 2017 [139]
Retrospective (subanalysis)	Nivolumab	HL	11 of 75 pts	30 (15–190)	Heterogeneous	BM and PBSC	3 of 11 pts.	1 of 11 pts.	10 of 11 pts alive	None	1 of 11 pts.	NA	Beköz et al. Ann Oncol. 2017 [147]
Retrospective	Nivolumab, Pembrolizumab, Ipilimumab	HL/NHL/MDS/AML	14 (N. HL = 10)	42 (18–231)	PT-Cy, CNI and MMF	BM (*n* = 12) and PBSC (*n* = 2)	6 of 14 pts	None	13 of 14 pts alive	2 pts (none with HL)	None	NA	Schoch et al. Blood adv. 2018 [148]
Retrospective	NA	HL	37 of 105 pts	51 (23–472)	PT-Cy, CNI and MMF	BM (*n* = 31), PBSC (*n* = 5), CB (*n* = 1)	33%	3%	94% at 3 years	4% at 3 years	6% at 3 years	NA	Paul et al. BBMT 2020 [142]
Retrospective	Nivolumab	HL/NHL/MM	18	83 (34–154)	CNI/PT-Cy-CNI -MMF	NA	3 pts receiving CNI alone none of the pts receiving PT-Cy	None	11 of 18	3 of 18 pts	5 of 18 pts	Circulating nivolumab found in plasma for up to 56 days after allo-HCT and binding PD-1 on T cells inducing T cell activation. Ratio T-reg/CD8: reduced in CNI group and increased in PT-Cy group	Nieto et al. Leukemia 2020 [140]
Retrospective (subanalysis)	Nivolumab	HL	39 of 74	58 (15–173)	Heterogeneous	BM (*n* = 2), PBSC (*n* = 37)	33%	35%	72% at 2 years	11%	13%	NA	Martinez et al. [141]
Retrospective	Nivolumab	HL	9	44 (27–100)	Heterogeneous	NA	8 of 9 pts	3 of 9 pts	8 of 9 pts	1 pt in SD	1 of 9 pts	NA	El Cheikh et al. BMT 2017 [149]
Retrospective	Nivolumab and Pembrolizumab	HL	25	59 (23–539)	Heterogeneous	BM (*n* = 12), PBSC (*n* = 11), CB (*n* = 2)	47%	None	52% at 1 year	27% at 1 year	8%	NA	Ito et al. Int J Hem 2020 [150]

Abbreviations: R/R HL: relapsed/refractory Hodgkin Lymphoma; allo-HCT: allogeneic hematopoietic cell transplantation; BM: Bone marrow; PBSC: peripheral blood stem cell; CB: cord blood; GvHD: graft versus host disease; SD: stable disease; OS: overall survival; NRM: non relapse mortality; CNI: calcineurine inhibitors; MMF: mychophenolate mofetil.

Subsequently, the success of BV in R/R HL prompted the use of CD30 as CAR-T target in HL, given its tissue specificity and expression patterns in HRS. The first proof of concept came from Wang and colleagues who treated 18 R/R patients in the context of a Phase 1 trial, demonstrating an ORR of 39% with 28% of patients showing stable disease at two months after CAR T cell infusion with a median PFS of 6 months. Tolerance was acceptable, with no severe cytokine release syndrome (CRS) or neurotoxicity events. Grade 1 or 2 CRS was instead seen in all patients, in most of cases resolutive without intervention [151]. Another phase I trial in R/R HL was conducted by Ramos and colleagues and included 9 patients undergone anti-CD30 CAR-T infusions without previous lymphodepletion. ORR was 33% without any significant toxicity post-infusion, including CRS [152]. A recent phase I/II study included 41 patients who received > 6 lines of therapy, including BV and PD-1 blockade and underwent ASCT. Depletion regimen comported a combination of bendamustine-fludarabine or cyclophosphamide–fludarabine). ORR was 72% with CR of nearly 60% and 1-year PFS and OS of respectively, 41% and 94%. The most common adverse effect was cytopenias (grade 3/4 thrombocytopenia in 24% and grade 3/4 neutropenia in 10%). CRS was in all patients of grade 1 without requiring therapy; no neurotoxicity was observed [134]. Based on these promising results, a number of studies of anti-CD30 CAR-T in R/R HL and other CD30+ lymphoproliferative disorders are now ongoing. (NCT05208853, NCT04526834, NCT04268706, NCT03383965, NCT05352828, NCT02917083, NCT04526834, NCT03383965).

## 5. Age Related Considerations

Despite the specific epidemiological pattern of HL, with a bimodal peak touching both young adults and elderly subjects, the optimal treatment for older patients still remains a matter of debate especially in the context of R/R disease, whereby outcomes have traditionally been poor because of poor tolerance to standard chemotherapy.

Although most of the trials did not preclude the access to immunotherapy and new targeted agents to elderly patients, combinatorial regimens and more intensive and sequential strategies are generally discouraged after 60 years. In this optic the place of PD-1 inhibitors in monotherapy [109]. as well as BV in combination with bendamustine [91,92,93,94,95] or less intensive polychemotherapy strategies, such as associations of gemcitabine, vinorelbine, and pegylated liposomal doxorubicin (GVD), constitute a valid opportunity for unfit patients who could not benefit from transplant strategies [153]. Nevertheless, prospective studies are needed for more stable recommendations in this particularly risky category of patients.

## 6. Conclusive Remarks

While for more than forty years the therapeutic scenario of HL remained mostly unchanged, dominated by very few, even if highly effective, available regimens, in this last decade we made the transition to a flourishing variety of agents targeting different actors of the immune system, together with HRS, not only in the context of R/R HL but also in early phase of HL treatment. Many of the new therapeutic strategies are still under investigation, however, the deep understanding of molecular mechanisms of its disease pathogenesis and treatment failure processes provides a benchmark to study the efficacy of novel therapies. If the place of ASCT remains almost untouched, the role of allo-HCT is so far less harmonized across transplant centers and needs to be necessarily re-evaluated in the era of the other immunotherapeutic approaches. Importantly, outstanding open questions remain the right timing and bridge-to-transplant regimens as well as the ideal GvHD prophylaxis to use in case of pre-transplant exposition of PD-1 blockade, with an emerging role of PT-Cy. Importantly, most of the data concerning the place of allo-HCT were generated in the era pre-BV and before immune checkpoint inhibitors. As a general concept, although lack of standardized guidelines, the tendency is to reserve allogeneic procedures to post-ASCT relapses after opportune disease control and possibly after a more or less extended PD-1 inhibitor washout [83] Given the success of monoclonal anti-CD30 therapeutics, adoptive T cell therapy is finally translating interesting phase I results into more advanced treatment protocols. Altogether, these advancements have completely changed the therapeutical landscape of this dismal category of patients, developing a horizon of useful less toxic immunomodulatory agents.

## Figures and Tables

**Figure 1 jcm-11-06574-f001:**
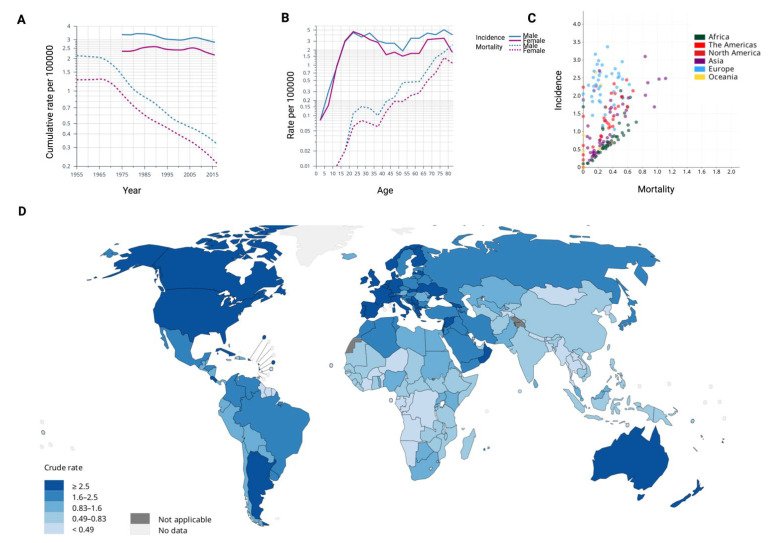
(**A**) Incidence and mortality of HL (cumulative rate per 100,000) grouped by sex across the time. Lines are smoothed by the locally weighted linear regression (LOESS) algorithm. Rates are shown on a semi-logarithmic scale. (**B**) Incidence and mortality of HL (rate per 100,000) by sex across ages. Rates are shown on a semi-logarithmic scale. (**C**) Relationship between incidence and mortality across countries (indicated as an age-standardized rate -ASR). (**D**) Map showing the estimated crude incidence rates in 2020, for HL in both sexes and across all ages. Data used to generate this figure were extracted from the Global Cancer Observatory project (International Agency for Research on Cancer).

**Figure 2 jcm-11-06574-f002:**
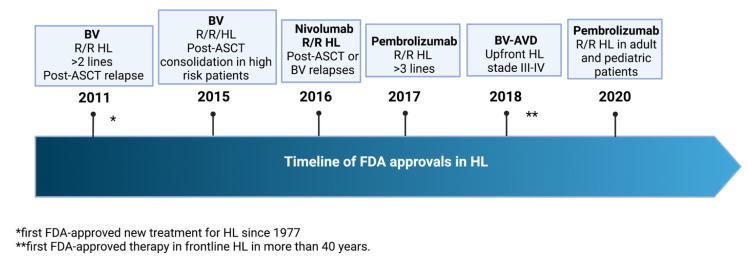
Timeline history of FDA approvals in Hodgkin lymphoma.

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
