# Peer review of "Filling the Gap: The Immune Therapeutic Armamentarium for Relapsed/Refractory Hodgkin Lymphoma"

_jcm, 2022, doi:10.3390/jcm11216574_

Round 1

Reviewer 1 Report

This paper by Dr. Hazane Leroyer and colleagues deals about one of most important questions in Hodgkin lymphoma, namely the immune-therapeutical armentarium in r/r HL (r/r HL patients represent an unmet medical need) . 

It is really a very useful paper because it depicts very well the different alternatives for the treatment of this heavily pretreated subset of HL patients and above all, the interest is the section about transplant, especially allogeneic HCT which remains one controversial subject. Indeed the recent literature dedicated to HL and transplant displayed alternatives donors like the haploidentical donor and the trials testing sequence BV and/or anti-PD1 inhibitor followed by allogeneic transplant. Here the reader can understand easily the rationale for each drug, or combination and the emerging strategies.

A few minor comments follow which the authors could consider in a subsequent revision including these comments:

-       The age consideration is mentioned in the first section, and it could be interesting to add the debate of upper limit for each strategy. One short section just before conclusion could help the readers to understand better the alternatives at each age category and the upper age limit for each strategy.

-       The section about pros/cons of a sequence anti-PD-1 and transplant and could be summarized in a table with references 

o   One row per trial and benefit in terms of outcome in one column and toxicities in the second column.  

-       Table 1 could be improved 

o   the name of the drug in the column 1 are often unproperly cut.

o   Sections of each category of drugs would be useful (Anti-PD1, combination anti-PD-1 + X,  CAR-T cells…)

o   Start dates in the sixth column looks in non-english format, necessary to check. 

o   The population of patient targeted in each study ; 1st relapse? Relapse after ASCT ? or all relapses whatever the number of previous lines received 

Author Response

Comments to the Author

This paper by Dr. Hazane Leroyer and colleagues deals about one of most important questions in Hodgkin lymphoma, namely the immune-therapeutical armentarium in r/r HL (r/r HL patients represent an unmet medical need) . 

 It is really a very useful paper because it depicts very well the different alternatives for the treatment of this heavily pretreated subset of HL patients and above all, the interest is the section about transplant, especially allogeneic HCT which remains one controversial subject. Indeed the recent literature dedicated to HL and transplant displayed alternatives donors like the haploidentical donor and the trials testing sequence BV and/or anti-PD1 inhibitor followed by allogeneic transplant. Here the reader can understand easily the rationale for each drug, or combination and the emerging strategies.

 We thank Reviewer #1 for this positive comment on our work.

A few minor comments follow which the authors could consider in a subsequent revision including these comments:

-       The age consideration is mentioned in the first section, and it could be interesting to add the debate of upper limit for each strategy. One short section just before conclusion could help the readers to understand better the alternatives at each age category and the upper age limit for each strategy.

We thank the Reviewer for this very pertinent comment. As suggested, we add a paragraph before the conclusions entitled: “Age-related considerations” to delve into this topic.

The paragraph reads as follows:

“Despite the specific epidemiological pattern of HL, with a bimodal peak touching both young adults and elderly subjects, the optimal treatment for older patients remains still matter of debate especially in the context of R/R disease, whereby outcomes have traditionally been poor because of poor tolerance to standard chemotherapy.

Although most of the trials did not preclude the access to immunotherapy and new targeted agents to elderly patients, combinatorial regimens and more intensive and sequential strategies are generally discouraged after 60 years. In this optic the place of PD1 inhibitors in monotherapy,109 as well as BV in combination with bendamustine91,92–95 or less intensive polychemotherapy strategies, such as associations of gemcitabine, vinorelbine, and pegylated liposomal doxorubicin (GVD), constitute a valid opportunity for unfit patients who could not benefit from transplant strategies.147 Nevertheless, prospective studies are needed for more stable recommendations in this particularly risky category of patients.”

-       The section about pros/cons of a sequence anti-PD-1 and transplant and could be summarized in a table with references

o   One row per trial and benefit in terms of outcome in one column and toxicities in the second column.  

We agree with the reviewer and we provided a new table (table two) in the revised version of the manuscript, summarizing the main characteristics and outcomes of studies investigating the use of anti-PD1 inhibition before transplant (including also translational findings where available).

-       Table 1 could be improved 

o   the name of the drug in column 1 are often unproperly cut.

We apologize for the inconvenience and we formatted it accordingly. Please note that we realized that this issue was related to the conversion to PDF from word through the submission system.

  • Sections of each category of drugs would be useful (Anti-PD1, combination anti-PD-1 + X,  CAR-T cells…)

This is an excellent suggestion and we integrated headers for each section.

  • Start dates in the sixth column looks in non-english format, necessary to check. 

We adjusted accordingly

o   The population of patient targeted in each study ; 1st relapse? Relapse after ASCT ? or all relapses whatever the number of previous lines received 

We integrated this information in the revised table 1

Reviewer 2 Report

Comments – major

1.     P7, lines 291-293. It is incorrect to state that “50% of the patients received ASCT without prior high dose chemotherapy…88” in the study by Chen et al. All 33 patients in this study who underwent autologous stem cell transplantation did so after receiving high dose chemotherapy, 20 after BEAM, 11 after CBV and 2 after BEAM plus yttrium-90. The authors seem to have been confused by the way pre-transplant treatment is described in the article by Chen, in which the high dose chemotherapy is bundled with the ASCT as a package. It is quite important that this error be corrected because autologous stem cell transplantation is always preceded by a high dose conditioning regimen, which, in fact, is the most effective component of the overall relapse treatment and contributes the most to curing the patient. If there were no high dose chemotherapy, there would be no need or use for the transplantation.

2.     Table 1. The odd truncation of NCT numbers, column head Phas   e and the start dates make reading this table confusing. This table would be more useful if this truncation were corrected.

3.     Table 1 is of very limited usefulness, providing no outcome data or results. Presently it is just a list of article titles. Much more useful would be a table showing numbers of patients studied, response rates and outcome data for each study.

4.     Table 1. At least one citation should be provided for each of the studies listed. Readers should be able to go from the table to specific citations to see the original data for themselves.

Comments – minor

1.     P1, line 41. “constituting” would be a better than “construing”, which seems out of place here.

2.     P2, line 80. “growth harshly” does not make sense. Perhaps the authors meant “rise steadily”.

3.     P4, line 158. “consistently” would be better than “constantly”.

4.     P5, line 222. Brentuximab vedotin should be spelled out fully the first time it is mentioned, after which the abbreviation BV would be acceptable.

5.     P7, line 276 and beyond. There is no need to capitalize brentuximab vedotin, nivolumab,  pembrolizumab or fludarabine. They are generic names.

6.     P7, line 318. The spelling of epitopes should be corrected.

7.     P14, line 555. “made the transition” would be better than “assisted”.

Author Response

Comments to the Author

P7, lines 291-293. It is incorrect to state that “50% of the patients received ASCT without prior high dose chemotherapy…88” in the study by Chen et al. All 33 patients in this study who underwent autologous stem cell transplantation did so after receiving high dose chemotherapy, 20 after BEAM, 11 after CBV and 2 after BEAM plus yttrium-90. The authors seem to have been confused by the way pre-transplant treatment is described in the article by Chen, in which the high dose chemotherapy is bundled with the ASCT as a package. It is quite important that this error be corrected because autologous stem cell transplantation is always preceded by a high dose conditioning regimen, which, in fact, is the most effective component of the overall relapse treatment and contributes the most to curing the patient. If there were no high dose chemotherapy, there would be no need or use for the transplantation.

We vividly thank the reviewer for highlighting this inaccuracy, due to a misleading wording. We better clarified this point in the revised version of the manuscript:

A phase II trial exploring its role as second line treatment before high dose chemotherapy and ASCT showed ORR over 60% with a complete remission rate of 35%. Importantly in this study, almost 50% of the patients were in PR or CR after BV alone without additional combination chemotherapy, before receiving intensification and ASCT, paving the way for the possibility of less toxic pre-transplant salvage regimens

Concerning table 1:

  • The odd truncation of NCT numbers, column head Phas e and the start dates make reading this table confusing. This table would be more useful if this truncation were corrected.

Done. Please note that we realized that this issue was related to the conversion to PDF from word through the submission system.

  • It is of very limited usefulness, providing no outcome data or results. Presently it is just a list of article titles. Much more useful would be a table showing numbers of patients studied, response rates and outcome data for each study.

We agree with the reviewer and we modified table 1 accordingly, including whenever possible, preliminary or definitive results, number of included patients and references.

  • At least one citation should be provided for each of the studies listed. Readers should be able to go from the table to specific citations to see the original data for themselves.

Done, see above.

Minor comments :

  1. P1, line 41. “constituting” would be a better than “construing”, which seems out of place here.

Done (line 48 of the revised manuscript)

  1. P2, line 80. “growth harshly” does not make sense. Perhaps the authors meant “rise steadily”.

Done

  1. P4, line 158. “consistently” would be better than “constantly”.

Done (line 150 of the revised manuscript)

  1. P5, line 222. Brentuximab vedotin should be spelled out fully the first time it is mentioned, after which the abbreviation BV would be acceptable.

Spelled at line 255 of the revised manuscript

  1. P7, line 276 and beyond. There is no need to capitalize brentuximab vedotin, nivolumab,  pembrolizumab or fludarabine. They are generic names.

Done

  1. P7, line 318. The spelling of epitopes should be corrected.

Done, line 292 of the revised manuscript

  1. P14, line 555. “made the transition” would be better than “assisted”.

Done, line 526 of the revised manuscript

Round 2

Reviewer 2 Report

No additional comments.